# *Arabidopsis Toxicos en Levadura 12* Modulates Salt Stress and ABA Responses in *Arabidopsis thaliana*

**DOI:** 10.3390/ijms23137290

**Published:** 2022-06-30

**Authors:** Feng Kong, Katrina M. Ramonell

**Affiliations:** 1Department of Biological Sciences, The University of Alabama, Tuscaloosa, AL 35401, USA; fkong@crimson.ua.edu or; 2Department of Plant Pathology, University of Georgia, Athens, GA 30602, USA

**Keywords:** salt tolerance, ABA, ROS, *Arabidopsis thaliana*

## Abstract

Salt is one of the most common abiotic stresses, causing ionic and osmotic pressure changes that affect plant growth and development. In this work, we present molecular and genetic evidence that *Arabidopsis Toxicos en Levadura 12 (ATL12)* is involved in both salt stress and in the abscisic acid response to this stress. We demonstrate that *ATL12* is highly induced in response to salt stress and that *atl12* mutants have a lower germination rate, decreased root length, and lower survival rate compared to the *Col-0* wild-type in response to salt stress. Overexpression of *ATL12* increases expression of the salt stress-associated genes *SOS1/2*, and ABA-responsive gene *RD29B*. Additionally, higher levels of reactive oxygen species are detected when *ATL12* is overexpressed, and qRT-PCR showed that *ATL12* is involved in the *AtRBOHD/F*-mediated signaling. *ATL12* expression is also highly induced by ABA treatment. Mutants of *atl12* are hypersensitive to ABA and have a shorter root length. A decrease in water loss and reduced stomatal aperture were also observed in *atl12* mutants in response to ABA. ABA-responsive genes *RD29B* and *RAB18* were downregulated in *atl12 mutants* but were upregulated in the overexpression line of *ATL12* in response to ABA. Taken together our results suggest that *ATL12* modulates the response to salt stress and is involved in the ABA signaling pathway in *Arabidopsis thaliana*.

## 1. Introduction

In nature, salt stress is a common abiotic stress that changes plants’ physiology and metabolism and adversely affects overall growth and crop production in agriculture [1]. More than 20% of the world’s arable lands are affected by high salt stress [2,3]. It is estimated that by the year 2050, up to half of the world’s arable land will be lost due to increased salinization [4]. Since the availability of arable lands to feed an increasing world population is limited; a clear understanding of the underlying signal transduction events in the plant response to salt stress is crucial for understanding both plant-environment interactions and in developing transgenic strategies to improve salt tolerance in crops.

To counter salt stress, plants have developed various mechanisms to respond and survive [5]. Due to the osmotic pressure and ionic toxicity caused by salt stress, maintaining ion homeostasis is important to survive under salt stress [6]. NaCl is the major salt form in soils and Na^+^ accumulation is the primary physiological change observed in cells in response to salt stress [7]. Excess Na^+^ usually causes the production of a variety of damaging oxygen species in a plant cell [8]. Plants have evolved complex mechanisms to exclude excess Na^+^ ions to preserve overall growth and survival [9]. The vacuolar-localized Na^+^/H^+^ transporter pathway and salt overly sensitive (SOS) pathway are two of the most studied mechanisms. The vacuolar-localized Na^+^/H^+^ transporter helps transfer excess Na^+^ from the cytoplasm to the vacuole, while the salt overly sensitive (SOS) pathway is a conserved regulatory system in plants that mediates active Na^+^ extrusion from the cytoplasm to the outside of the cell and maintains ion homeostasis in response to salt stress [10,11].

Reactive oxygen species (ROS) production is another early signaling change that occurs in response to salt stress [12]. ROS including hydrogen peroxide (H_2_O_2_), hydroxyl radicals (OH•), superoxide anions (O_2_^•−^), and singlet oxygen (^1^O_2_) is accumulated in plants in response to drought, salt, and cold stress [13]. Recent studies showed that ROS accumulation causes oxidative damage and even cell death if not cleared by the ROS scavenger system [14,15]. ROS at low levels is recognized as a signaling molecule involved in both normal growth [16] and in response to biotic and abiotic stresses [15,17]. The activity of ROS as second messengers is closely related to multiple signaling pathways, including Ca^2+^-related signaling [18], the Salt-Overly-Sensitive (SOS) signaling pathway [10], and plant hormone signaling pathways [19]. In general, reactive oxygen species production and changing intracellular Ca^2+^ levels will initiate a downstream protein phosphorylation cascade, including the induction of Calcium-dependent protein kinases (CPKs), that in turn target proteins directly involved in defense or transcription factors controlling the expression of specific salt stress-responsive genes [5,19,20]. The induced stress-responsive genes include some defense-related proteins such as the SOS protein, which extrudes excess ions to maintain ion homeostasis. SOS is also responsible for the generation of the phytohormone abscisic acid (ABA) that is synthesized under high salt conditions and mediates salt tolerance through an ABA-dependent pathway, that eventually promotes salt tolerance and coordinates plant growth and development [21]. In addition, increasing evidence has shown that large numbers of genes are involved in the recognition and response to salt stress in plants including several *Arabidopsis* receptor-like kinases (RLK) and receptor-like proteins (RLP) [22,23]. The *Arabidopsis thaliana* histidine kinase receptor protein HK1 is known to complement the loss of the yeast osmotic sensor Sln1 and is involved in response to high osmotic stress [24]. A receptor-like kinase in rice (*Oryza sativa*), Salt Intolerance 1 (SIT1), a lectin receptor-like kinase, has also been shown to mediate salt sensitivity [25]. One of the chitin receptors, *CHITIN ELICITOR RECEPTOR KINASE 1 (CERK1)*, that is involved in chitin-triggered innate immunity in Arabidopsis has also been shown to play a role in mediating salt signaling [26]. 

*The Arabidopsis Toxicos en Levadura (ATL)* gene family encodes a large group of *RING* (REALLY-INTERESTING-NEW-GENE) zinc-finger domain-containing proteins, which are highly conserved [27,28]. Previous studies have shown that ATL family proteins are involved in multiple defense pathways in *Arabidopsis thaliana* that are induced by fungal infection, bacterial invasion, wounding stress, cold, drought, and salt stress [29]. For example, the *Arabidopsis* RING E3 ubiquitin ligase *AtATL80* is known to be involved in cold stress response in sufficient phosphate growth conditions [30]. The *Arabidopsis* E3 ubiquitin ligase *ATL9* is involved in fungal defense [31], and suppression of *AtATL78* increases tolerance to cold stress and decreases tolerance to drought stress in Arabidopsis thaliana [32]. Our previous studies have shown that *ATL12*, a gene belonging to the *Arabidopsis Toxicos en Levadura 2* family, is involved in crosstalk between hormonal, chitin-induced, and NADPH oxidase-mediated defense responses in *Arabidopsis* [33].

In this study, we show that *ATL12* positively modulates the salt stress response in *Arabidopsis thaliana*. T-DNA insertional mutants of *atl12* showed decreased seed germination and root length growth, as well as a lower overall survival rate. *ATL12* is highly induced by salt stress and its expression is upregulated at late stages in response to NaCl stress. In addition, over-expression of *ATL12* increases the expression of the salt stress-associated genes SALT-OVERLY-SENSITIVE 1 and 2 (*SOS1* and *SOS2*), the ABA-responsive gene RESPONSIVE TO DESICCATION 29B (*RD29B*), and the chitin-receptor gene *CERK1*. An elevated oxidative burst is detected in over-expression of *ATL12* in response to NaCl stress, and qRT-PCR showed that *Arabidopsis thaliana* RESPIRATORY BURST OXIDASE HOMOLOGUE D/F (*AtRBOHD/F*) may be responsible for the ROS generation, suggesting that *ATL12* may positively regulate salt tolerance via regulating *AtRBOHD/F* mediated ROS production. Mutants of *atl12* plants are hypersensitive to ABA treatment, showing a decrease in water loss and reduced stomatal aperture in response to ABA. In contrast, *ATL12* overexpression (*OE-ATL12*) plants have a reduced sensitivity to ABA and show increased water loss and enlarged stomatal aperture compared to Col-0 wild type (WT). qRT-PCR showed expression of *ATL12* is highly induced by ABA treatment, and over-expression of *ATL12* upregulates the expression of ABA-responsive genes in response to salt, suggesting that *ATL12* may enhance salt tolerance through an ABA-mediated pathway in *Arabidopsis thaliana*. Together these results suggest that *ATL12* modulates the response to salt stress and is involved with an ABA-dependent pathway in *Arabidopsis thaliana*.

## 2. Results

### 2.1. ATL12 Positively Regulates Salt Tolerance in Arabidopsis Seedlings

To characterize *ATL12*, we first analyzed the *ATL12* gene structure and domain information. *ATL12* belongs to the *ATL2* gene family and encodes a 390 amino acid C3HC4 RING/Ubox protein with a signal peptide (red color) near the N-terminal region, a transmembrane domain (blue color), and a RING-finger domain (purple color) near the C-terminal region, as shown in Figure 1A. To determine if the ATL12 protein has homologs in other plant species, homology searches were performed. Selected ATL12 protein homologs were aligned and the result is shown in Appendix A. Since ATL12 encodes a protein containing a conserved RING domain, which is essential for E3 ubiquitin ligase activity, we also determined if the RING domain was conserved among the other ATL12 homologs. As shown in Figure 1A, all ATL12 homologs shared a RING domain. Phylogenetic trees were also generated to determine the evolutionary relationship between the genes. As shown in Figure 1B, the closest homolog of ATL12 in *Arabidopsis thaliana* (NM_127561) is the putative RING-H2 finger protein ATL12 (*LOC9320018*) in *Arabidopsis lyrata subsp. lyrata* with accession number XM_002885958. Another *Arabidopsis thaliana* RING/U-box superfamily protein ATL42 (AT4G28890) is also a close homolog of ATL12 in *Arabidopsis thaliana*. Other close homologs of *ATL12* are found in *Brassica napus* (XM 013860) and *Brassica rapa* (XM_009130924). In addition, several proteins in other plant species share similarities with ATL12, including in Melon (*Cucumis melo*), soybean (*Glycine max*), maize (*Zea mays*), and rice (*Oryza sativa*).

In *Arabidopsis*, there are over 61 U-box domain-containing proteins with predicted E3 ubiquitin ligase activity and over 450 predicted proteins that contain one or more RING domains [34]. E3 ubiquitin ligases are known to be involved in plant defense against both biotic and abiotic stress [35]. Since *ATL12* encodes a protein containing a conserved RING domain, which is essential for E3 ubiquitin ligase activity, we confirmed ATL12’s E3 ubiquitin ligase activity using an in vitro ubiquitination assay, as shown in Figure 1C. To perform the in vitro ubiquitination assay, we expressed and purified *Arabidopsis* ATL12, UBC8 (E2), UBQ14 (Ub), and UBA2 (E1) in *E. coli*. As shown in Figure 1C, we noted that proteins were ubiquitinated only in the presence of E1 (UBA2), E2 (UBC8), and 6× His tagged ubiquitin, demonstrating that ATL12 has E3 ubiquitin ligase activity.

To analyze the phenotypes of *atl12* mutants in response to salt stress, we determined if the sensitivity of the mutants to salt was altered when challenged with high salt. Homozygous T-DNA insertional mutants of *atl12* and a transgenic line overexpressing *ATL12* from a previous study were used in these experiments [33]. Quantitative RT-PCR was performed to confirm the expression of the *ATL12* mRNA transcript in different seedlings, as shown in Figure 2A. The qRT-PCR results indicated that expression of *ATL12* is significantly decreased in all three T-DNA insertional mutants of *atl12* compared to *Col-0* WT, while *ATL12* was overexpressed in the overexpression (OE) line. We then assessed the salt tolerance of *ATL12* during seed germination and in the early stages of plant development. Seeds of Col-0 WT, the three *atl12* mutants, and the *ATL12* OE line were planted on agar plates containing 150 mM NaCl and the germination rates were observed after seven days (Figure 2B). Under normal conditions, germination rates are not significantly different among Col-0 wild type, *atl12* mutants, and the *ATL12* overexpression lines. When the plants were treated with 150 mM NaCl, the germination rate of the *atl12* mutants was significantly decreased compared to the Col-0 wild type and the germination rate in the OE *ATL12* line increased (Figure 2D). To study the function of *ATL12* in response to salt stress in more detail, we also determined if there were changes in the primary root growth of Col-0 wild type, *atl12* mutants, and OE *ATL12* under salt stress conditions (Figure 2C). Under 150 mM NaCl conditions, *atl12* mutants were hypersensitive to salt treatment and had decreased root length growth compared to *Col-0* WT. Interestingly, there was no significant change in root length growth observed in the OE *ATL12* line compared to WT (Figure 2E). To further investigate the role of *ATL12* in salt stress, we also determined if mutations in *ATL12* altered the plant phenotype in post-germination stages in response to high salt treatment. Four-week-old seedlings of *Col-0*, *atl12* mutants, and the OE *ATL12* line were treated with 200 mM NaCl and leaf appearance was observed after seven days (Figure 2F,G). The results showed that the *atl12* mutants are more sensitive to salt stress compared to Col-0 WT and exhibited more cell death on the leaves. Taken together these results indicate that *ATL12* plays a positive role in salt tolerance.

### 2.2. Expression Pattern of ATL12 under High Salt Conditions

To analyze the expression pattern of *ATL12* at the tissue level, we fused the promoter of *ATL12* with the GUS reporter gene and generated transgenic plants. As shown in Figure 3A, *ATL12* is expressed in the root, stem, and leaves of two-week-old seedlings, and in mature flowers. To confirm the tissue-specific expression, the expression levels of *ATL12* were tested using RT-PCR in the root, leaf, and stem of two-week-old seedlings, in mature flowers, and in dried seeds of Col-0 wild-type plants. As shown in Figure 3B, the *ATL12* transcript was detected in the root, stem, and flower, but its expression was relatively low in the leaf tissue.

To determine if the expression of *ATL12* is induced by salt treatment, we used histochemical staining of *ATL12* promoter GUS fusion transgenic plants treated with 150 mM NaCl. The results showed that *ATL12* is rarely expressed under normal conditions, while p*ATL12*:GUS expression was strongly induced after being treated with 150 mM NaCl for 16 h, as shown in Figure 3C. Additionally, we performed qRT-PCR to examine the expression of *ATL12* in *Arabidopsis* treated with 150 mM NaCl at early (1 h, 2 h, and 4 h) and late stages (16 h and 24 h) post-salt treatment. The salt stress-responsive genes salt-overly sensitively 1 and 2 (*SOS1* and *SOS2*) and the reference gene *beta-actin* were also determined using qRT-PCR. As shown in Figure 3D, the expression of *ATL12* is increased after 4 h and is significantly increased at 16 h. Compared to *ATL12*, *SOS1* and *SOS2* are significantly increased during the early stages of salt stress, as shown in Figure 3D. Our results show that *ATL12* expression is salt-inducible and its expression is upregulated at later stages in response to NaCl.

### 2.3. Overexpression of ATL12 Increases the Expression of Salt Stress-Associated Genes

We noticed that *ATL12* expression is upregulated at later stages during salt stress (Figure 3D) and that over-expression of *ATL12* increases seed germination rate (Figure 2). To better understand the function of *ATL12* in response to salt stress, we determined if over-expression of *ATL12* alters the expression of other well-characterized salt stress-associated genes. Two-week-old *Col-0* wild-type plants, *atl12*, and OE lines of *ATL12* were treated with 150 mM NaCl for 16 h. Distilled water treatment was used as a mock control. Gene expression of *ATL12*, salt-responsive genes *SOS1* (SALT-OVERLY-SENSITIVE 1) and *SOS2* (SALT-OVERLY-SENSITIVE 2), ABA-responsive gene *RD29B*, and chitin-responsive gene *CERK1* were then determined. As shown in Figure 4A, under normal conditions, there are no significant changes in the gene expression of *SOS1*, *SOS2*, *CERK1*, and *RD29B*. The transcript level of *ATL12* is downregulated in the *atl12* mutant and is upregulated in the OE *ATL12* line in the water controls. Under high salt conditions, gene expression of *SOS1* and *SOS2* was significantly increased in the OE line of *ATL12* compared to wild-type plants, while *SOS1* and *SOS2* expression levels were decreased in the *atl12* mutant, as shown in Figure 4B. Together these results suggest that *ATL12* may be involved in plant salt responses via regulation of *SOS* signaling in order to mediate Na^+^ ion extrusion in response to salt stress. In addition, we noticed that the ABA-responsive gene *RD29B*, and the chitin-responsive gene *CERK1* showed increased expression in the ATL12 overexpression line, whereas *CERK1* expression was downregulated in the *atl12* mutant, suggesting that *ATL12* modulates salt stress through multiple signaling events and *ATL12* may play a broader role in abiotic stress responses.

### 2.4. Detection of Reactive Oxygen Species Production in Over-Expression of ATL12 in Response to NaCl Stress

In plants, NADPH oxidase-mediated reactive oxygen species (ROS) generation has been found to play an important role in salt stress responses [36,37]. Our previous work showed that *ATL12* is involved in AtRBOHD/F-mediated ROS production in response to the fungal pathogen *Golovinomyces cichoracearum* (33). To determine if *ATL12* and ROS generation were also involved in the response to salt stress, we performed DAB (3,3′-diaminobenzidine) staining to detect H_2_O_2_ production. Four-week-old *Col-0* wild-type, *atl12* mutants, and OE-*ATL12* leaves were treated with 150 mM NaCl for 16 h, and no treatment (NT) was used as a control. The leaves were then stained with DAB (3,3′-diaminobenzidine) staining solution. As Figure 5A shows, compared to wild-type *Col-0*, the leaves of OE *ATL12* showed more ROS generation in response to salt stress, while *atl12* mutants exhibited less ROS generation. This suggests that *ATL12* may be associated with ROS generation in response to salt stress. 

It has been shown that the accumulation of ROS in response to salt stress can have toxic effects in plants, leading to programmed cell death [12]. In our experiment (Figure 5), we noted that ROS accumulated in the OE-*ATL12* line in response to salt stress (Figure 5A). In order to determine if *ATL12*-mediated ROS production is inducing cell death in response to high salt conditions, we used trypan blue staining to observe areas of cell death in the leaves of plants under 150 mM salt conditions. Detached leaves from four-week-old *Col-0* wild-type, *atl12* T-DNA insertional mutants, and the OE *ATL12* line were treated with NaCl for 16 h and distilled water treatment was used as the control. As shown in Figure 5B, there is no significant difference between *Col-0* wild-type, the *atl12* mutants, and the OE *ATL12* line in the levels of cell death observed in the leaves under normal conditions. In the salt treatment, there is increased cell death in the leaves of all the plants tested compared to normal conditions. However, no significant differences were observed between the salt-treated *Col-0* wild-type, the *atl12* mutants, and OE-*ATL12*, suggesting that *ATL12*-mediated ROS production may not lead to cell death under salt stress. Other than its toxic effects, ROS are also known to function as signaling molecules in response to salt stress [12,37]. In Arabidopsis, the multiple respiratory burst oxidase homolog (RBOH) proteins have been suggested to be involved in the regulation of salt stress responses [38]. The expression of two respiratory burst oxidase homologs, *AtRBOHD* and *AtRBOHF* are known to be highly induced under salinity stress and are responsible for ROS-mediated ion homeostasis [36,37]. To confirm if *AtRBOHD* and *AtRBOHF* were involved in the salt stress response and if their expression was induced by salt stress, a germination rate assay and a root length assay were performed using *Col-0* WT and *atrbohd* and *atrbohf* mutants in response to 150 mM NaCl, as shown in Figure 5C–F. Mutants of *atrbohd* and *atrbohf* showed lower germination rates compared to the *Col-0* wild-type. Only mutants of *atrbohd* showed a decrease in root length growth compared to wild-type plants, suggesting that *AtRBOHD* and *AtRBOHF* are involved in the salt stress response. In addition, we used qRT-PCR to determine the *AtRBOHD* and *AtRBOHF* expression patterns in response to salt stress. Two-week-old Col-0 wild-type plants were treated with 150 mM NaCl for 1 h, 2 h, 4 h, 16 h, and 24 h, and expression levels of *ATL12*, *AtRBOHD*, and *AtRBOHF* were evaluated. As shown in Figure 5G, *AtRBOHD* expression is highly induced at 1 h, 4 h, and 16 h, while *AtRBOHF* expression is induced at 4 h and 16 h. These results confirmed that *AtRBOHD/F* are involved in the Arabidopsis response to salt stress.

To further analyze if mutations in *ATL12* and its overexpression alter *AtRBOHD/F* expression, two-week-old *Col-0* wild-type, *atl12* mutants, and the OE *ATL12* line were treated with 150 mM NaCl for 16 h, and gene expression was evaluated using qRT-PCR. The results are shown in Figure 5H,I. Under normal conditions (water control), there is no significant change in the *AtRBOHD/F* expression. Under salt stress, we noted that *AtRBOHD/F* expression was upregulated in the OE *ATL12* line, while their expression was downregulated in the *atl12* mutant. Taken together, these data suggest that ATL12 may play a central role in AtRBOHD/F-mediated ROS generation in response to salt stress.

### 2.5. ATL12 Expression Is ABA Inducible and Negatively Regulate ABA-Dependent Response

Abscisic acid signaling is known for regulating multiple physiological processes and is involved in abiotic stress tolerance [39]. Our results show that the ABA-responsive *RD29B* gene’s expression is induced in the OE-*ATL12* line in response to salt stress (Figure 4B), which suggests that *ATL12* might be involved in salt stress responses via ABA signaling. To investigate whether *ATL12* expression is induced by ABA, *ATL12*’s expression pattern in response to ABA treatment was examined via qRT-PCR. Two-week-old Col-0 wild-type plants were treated with 0.75 μM ABA and tissues were harvested at different time points (Early-stage: 0 h, 1 h, 2 h, 4 h, and 8 h; Late-stage: 16 h and 24 h). Expression of the ABA-responsive gene *RD29B* was also examined as a marker gene and the *ACT2* gene was used as an internal control. As shown in Figure 6A, *ATL12* expression is continuously induced at all the time points and the expression of *RD29B* is highly induced at all time points. To further dissect the potential interaction between *ATL12* and ABA, *Col-0* wild-type, *atl12* mutants, and OE *ATL12* plants were treated with 100 mM ABA for 2 h, and then gene expression of *ATL12*, ABA-inducible genes *RD29B* and *RAB18*, and *ACT2* was evaluated by qRT-PCR. There were no significant changes in *RD29B* and *RAB18* expression under normal conditions (water treatment) as shown in Figure 6B. It was noted that *RD29B* and *RAB18* expression were upregulated in OE-*ATL12* plants, while their expression was downregulated in the *atl12* mutant, as shown in Figure 6C. These data suggest that *ATL12* gene expression is highly induced by ABA and that *ATL12* is involved in some way with the ABA signaling pathway.

To investigate if *atl12* and over-expression of *ATL12* alter the sensitivity of plants to ABA, the germination rate and root length of Col-0 WT, *atl12*, and OE-*ATL12* were tested under normal (water), and ABA treated conditions. As shown in Figure 7A,B, there were no significant differences in the germination rate between the *Col-0*, *atl12*, and *OE-ATL12* seedlings under both the normal conditions and ABA treatment. In the root length assay, no significant difference was also observed between *Col-0*, *atl12*, and *OE-ATL12* line under normal conditions as shown in Figure 7C,D. However, the *atl12* mutants showed significantly inhibited root length growth compared to *Col-0* WT (Figure 7D) and *OE-ATL12* showed significantly increased root length with ABA treatment. These results suggest that *ATL12* negatively influences ABA inhibition of root length growth. 

Since ABA is known for regulating water loss and stomatal aperture [40,41], and to further characterize the role of *ATL12* in ABA responses, we investigated if *atl12* plants had altered stomatal aperture in response to ABA. Fresh leaves from four-week-old plants were immersed in a stomatal opening solution and then treated with ABA or distilled water (control). Images of leaf stomata were then obtained via scanning electron microscopy and the stomatal aperture index was analyzed (Figure 7E). As shown in Figure 7F,G, under normal conditions, there was no significant difference in the stomatal aperture. However, after ABA treatment, mutants in *atl12* have a reduced stomatal aperture, while leaves from OE-*ATL12* plants have a significantly larger stomatal aperture when compared to Col-0 WT. Taken together, these results suggest that *ATL12* expression is ABA inducible and negatively regulates ABA-dependent responses. We also evaluated if mutations in ATL12 alter the plant’s ability to prevent water loss in response to ABA. Detached leaves from four-week-old plants were incubated in 100 μM ABA for 2 h and then the leaf fresh weight was measured every hour for four hours. As shown in Figure 7H,I, under normal conditions, there was no significant difference in water loss rates between the *atl12* mutant, *Col-0* WT, and the OE line. However, the OE-*ATL12* line lost more water compared to *Col-0* WT in response to ABA treatment. Mutants in *atl12* have decreased water loss compared to *Col-0* WT suggesting that *ATL12* negatively influences ABA-mediated water loss.

## 3. Discussion

### Arabidopsis Toxicos en Levadura 12, Modulates Salt Stress Responses through Multiple Signaling Events

Salt is one of the most common abiotic stresses that cause changes in ionic and osmotic pressure, affecting plant growth and development [2]. However, the cellular and molecular mechanisms behind how plants regulate salt stress largely remain unclear [1]. In this study, we characterize the gene *ATL12*, which belongs to the *Arabidopsis Toxicos en Levadura 2 (ATL2)* gene family and encodes a conserved RING-Zinc finger protein with E3 ubiquitin ligase activity. RT-PCR showed that *ATL12* is continually expressed in roots, leaves, stems, and flowers and T-DNA insertional mutants of *atl12* showed decreased seed germination rates and reduced root length compared to Col-0 WT. Over-expression of *ATL12* resulted in an increased germination rate compared to WT. In addition, *ATL12* was found to be highly induced by salt stress, and its expression is upregulated at late stages in response to NaCl stress. These results strongly suggest that *ATL12* plays a positive role in the plant response to salt stress.

The Salt Overly Sensitive (SOS) signaling pathway, including the three major proteins SOS1, SOS2, and SOS3, mediate active Na^+^ extrusion at the cell membrane and maintain ion homeostasis in response to salt stress, which is believed to play an important role in plant salt tolerance [11]. Our study showed that *SOS1* and *SOS2* are upregulated in the overexpression line of *ATL12* in response to salt, which suggests that *ATL12* may be involved in Na^+^ ion extrusion via SOS signaling in order to assist plant cells in avoiding the toxic effects of excess Na^+^ accumulation. Further experiments to investigate Na^+^/K^+^ ion accumulation in *Col-0*, *atl12* mutants, and *OE-ATL12* after salt treatment would be valuable in order to confirm this hypothesis. In addition, ROS function as secondary signaling components in stress responses [17]. Excess ROS production in plants causes cellular damage, however, controlled levels of ROS play important roles in both biotic and abiotic stress tolerance [36,38]. A previous study has also shown that the *NADPH* oxidase genes *AtRBOHD* and *AtRBOHF* are involved in ROS-dependent regulation of Na^+^/K^+^ homeostasis in Arabidopsis under salt stress [37]. In this study, we detected an elevated oxidative burst in the OE-*ATL12* in response to NaCl stress, while the *atl12* mutant generated less ROS compared to *Col-0* WT. These data suggest that *ATL12* may be involved in ROS signaling in response to salt stress. ROS is mainly produced by the respiratory burst oxidase homolog D and F (*RBOHD/F*) proteins in response to abiotic stress in Arabidopsis thaliana. Our results show that the expression of *AtRBOHD* and *AtRBOHF* are highly induced at 4 h post salt treatment. Over-expression of *ATL12* also upregulates the *AtRBOHD/F* expression, while *AtRBOHD/F* expression was downregulated in the mutant of *atl12*. Together these results suggest that *ATL12* may be involved in the SOS1/2-mediated Na^+^ homeostasis and *AtRBOHD/F* mediated ROS generation in response to salt stress. Further studies focusing on the crosstalk between SOS1/2-mediated ion homeostasis and *AtRBOHD/F* mediated ROS signaling will be important to understand the role of *ATL12* in mediating ion homeostasis under salt stress. *RD29B* (RESPONSIVE TO DESICCATION 29B) serves as a positive regulator in response to high salt stress through the ABA-activated signaling pathway [42]. Our results indicated that *RD29B* expression was upregulated in the over-expression line of *ATL12* in response to salt stress, suggesting that *ATL12* may modulate salt stress responses through the ABA signaling pathway. We also noted that *RD29B* and *RAB18* expression were upregulated in the over-expression line of *ATL12* in response to ABA. Compared to Col-0 WT, mutants of *atl12* displayed different phenotypes in their germination rate, root length growth, leaf water loss, and stomatal aperture in response to ABA. These data confirmed that *ATL12* is involved with the ABA signaling pathway. However, the detailed regulatory mechanisms in ABA-mediated responses to salt stress remain unclear and require further study.

*CERK1* (Chitin elicitor receptor kinase 1) is a major receptor for plants in response to fungal pathogens or chitin treatment [43]. Our work has shown that *ATL12* expression is highly induced by chitin treatment and that *ATL12* expression is reduced in the *cerk1* mutant after treatment with chitin [33]. A previous study also showed that *CERK1* linked both salt stress and the chitin immune response in Arabidopsis and is capable of mediating salt signaling [26]. The *cerk1* mutant is more susceptible to NaCl than wild-type plants and *CERK1* is necessary for NaCl-induced cytosolic calcium([Ca^2+^] cyt) increase, which induces a mechanism for salt tolerance [26]. However, little is known about *CERK1*’s role in salt stress. In the current study, we showed that *CERK1* expression is highly induced 2 h, 4 h, 16 h, and 24 h after salt treatment (Appendix A). *CERK1* expression is also upregulated in the over-expression line of *ATL12* compared to Col-0 WT, while its expression is downregulated in mutants of *atl12*, as shown in Figure 4. Together these results strongly suggest that *ATL12* may be involved in plant salt responses through *CERK1* signaling. Further study is required to fully characterize the relationship between *ATL12* and *CERK1* in response to salt stress. Considering all of our data, we propose a putative model for *ATL12*’s function in salt stress response, shown in Figure 8. *ATL12* is induced by NaCl stress after 4 h. *ATL12* activates the *AtRBOHD/F*-mediated ROS signaling, *SOS1/2* signaling, and ABA-dependent signaling pathway to induce the expression of a salt-responsive gene, which eventually increases the tolerance to salt stress.

In the current work, we present strong evidence demonstrating that *ATL12* modulates salt stress responses via *AtRBOHD/F*-mediated ROS signaling, *SOS1/2* signaling, and ABA-dependent signaling in Arabidopsis thaliana. Our results clearly indicate that *ATL12* has an important role in modulating salt stress in Arabidopsis thaliana via multiple signaling pathways and has a broad role in salt responses. Further work characterizing the precise interactions between ATL12 and these signaling pathways will be important in determining the overlapping processes that regulate plant responses to salt stress in the environment.

## 4. Materials and Methods

### 4.1. Plant Materials and Growth Conditions

*Arabidopsis thaliana* ecotype Columbia (Col-0) plants were obtained from our laboratory stock. T-DNA insertion mutants of *atl12* with Col-0 background (*SALK_21056C*, *SALK_066923C*, and *SALK_0950303C*) were obtained from the Arabidopsis Biological Resource Center (ABRC, Ohio State University, Columbus, OH, USA) [33]. All the plants were grown in the growth chamber under controlled conditions at 22 °C day/19 °C night with 16 h of light per 24 h and 50% humidity.

### 4.2. Bioinformatics Analysis and Phylogenetic Analysis

The complete sequences of *ATL12* were obtained from GenBank and TAIR (The *Arabidopsis* Information Resource (TAIR). Open-reading frames (ORFs) of *ATL12* were analyzed with ORF Finder (http://www.ncbi.nlm.nih.gov/gorf/orFig.cgi) (accessed on 15 October 2021), and its gene structure and functional domains were predicted with Uniprot (https://www.uniprot.org) (accessed on 15 October 2021) and Smart software (http://smart.embl-heidelberg.de/) (accessed on 15 October 2021). Tools used for general bioinformatics analysis and protein-specific domains were: http://www.ncbi.nlm.nih.gov/ and http://www.ebi.ac.uk/tools/ (accessed on 15 October 2021). Protein homology searches were performed with the Gramene program (https://www.gramene.org/#) (accessed on 15 October 2021) and the Phytozome program (https://phytozome-next.jgi.doe.gov; accessed on 10 January 2022). Selected nucleotide sequences were obtained from NCBI and aligned using MUSCLE with default parameter settings (accessed on 10 January 2022). Selected amino acid sequences were also aligned using Clustal Omega software (https://www.ebi.ac.uk/Tools/msa/clustalo/) (accessed on 10 January 2022) and highlighted in Boxshade (https://embnet.vital-it.ch/software/BOX_form.html; accessed on 10 January 2022). The alignment is available as Appendix A. Phylogenetic trees were generated using the neighbor-joining method in MEGAX (https://www.megasoftware.net; accessed on 11 January 2022) and bootstrap analysis with 1000 replicates. Branch length indicates divergence distance. Numbers on the branches designate percentage bootstrap support.

### 4.3. Generation of Transgenic Plants and Constructs

To determine if the over-expression of *ATL12* increases the tolerance to salt stress, *ATL12* sequence was amplified using Phusion^®^ High-Fidelity DNA Polymerase from New England Bio-Labs. For Gateway entry cloning, Taq DNA polymerase is added to the raw PCR product to generate 3′ A-Overhangs. The resulting PCR products are purified, and entry clones are generated by recombination into the pCR™8/GW/TOPO vector, using the pCRTM8/GW/TOPO^®^ TA Cloning^®^ Kit. To generate the overexpression constructs for *ATL12*, we utilized the destination vector pMDC 32 (35S promoter-attR1-CmR-ccdB-attR2). After sequencing the plasmid to confirm that the clone is in frame and oriented correctly, the recombinant construct is transformed into *Agrobacterium tumefaciens* strain (pGV3101) via chemical transformation. The floral dip method was used to generate overexpression transgenic plants and select the transformed resistant plants using kanamycin plates. The T3 progeny from these transformants will be used in future experiments. Transcript levels of *ATL12* were measured via quantitative RT-PCR in the over-expression lines to confirm our results.

### 4.4. Salt and ABA Tolerance Test

To assess the susceptibility of *Arabidopsis atl12* mutants to salt stress, the germination assay, and root length growth assay were performed. *Arabidopsis* seeds of *Col-0* WT, and *atl12* T-DNA insertional mutants were sterilized and then placed in the cold room for three days. For germination rate assay, over 36 seeds of wild-type, *atl12* mutants, and over-expression lines were then planted on 0.5× MS agar plates containing 150 mM NaCl and normal 0.5× MS agar plates to grow. The emergence of the radicle is used as germinated seed and the percentage of germination rates for the seedlings was then observed and calculated after 7 days. For root elongation assay, seeds of wild-type, *atl12* mutants, and overexpression seedlings were sterilized and dark-treated for 2 days and were grown on vertical MS medium plates for 2 days. After that, over 10 seeds with similar root lengths were transferred to MS medium supplemented with 150 mM NaCl and no salt for 7 days. The root length was measured by a ruler. The whole experiment was repeated five times. Statistical significance among samples was analyzed using one-way ANOVA followed by post-Hoc tests.

### 4.5. Histochemical Staining Assay

Transgenic *Arabidopsis* plants expressing p*ATL12*-GUS were generated and a histochemical staining assay was used to figure out the p*ATL12*-GUS (β-Glucuronidase) expression pattern. Primers *pATL12-ATL12* 5′-ATCCACCTTCATAAGCTGGTAATAGA-3′(forward) and *pATL12-ATL12* 5′-TGTTTTAGGATGGTGATTCGATGAG-3′(reverse) were used to generate the *ATL12* promoter region. p*ATL12* sequence was amplified using Phusion^®^ High-Fidelity DNA Polymerase from New England Bio-Labs. Taq DNA polymerase was added to the raw PCR product to generate 3′ A-overhangs and the resulting PCR products were purified and entry clones were generated by recombination into the pCR™8/GW/TOPO vector. The construct was sequenced and then switched onto destination vector CD3-754 (-attR1-CmR-ccdB-attR2-GUS) via LR reaction. Then the recombinant construct was transformed into *Agrobacterium tumefaciens* strain (pGV3101) via chemical transformation. The floral dip method was used to generate transgenic plants and selection for the transgenics was performed using kanamycin plates. The T3 progeny from these transformants was used in future experiments. The screened p*ATL12*-GUS seeds were placed into 1× MS liquid culture and grown to specific developmental stages. The seedlings and tissues were then collected and stained overnight at 37 °C in GUS staining buffer. Samples were de-stained for up to 8 h in 100% Ethanol and the GUS expression is observed directly under the dissecting microscope.

### 4.6. Reactive Oxygen Species (ROS) Detection via DAB Staining Assay

*Arabidopsis* plants *Col-0*, T-DNA insertional mutants of *atl12* and *ATL12* over-expression lines were grown under normal conditions (22 °C day/19 °C night with 12 h of light per 24 h and 50% relative humidity) for 3 weeks. Plants were then treated with 150 mM NaCl for 16 h, then at least 5 leaves were removed from the plant and placed in a 24-well microtiter plate. 1 mL of the 10 mM Na_2_HPO_4_ DAB staining solution (50 mg DAB, 45 mL sterile H_2_O, 25 μL Tween 20 (0.05% *v*/*v*) and 2.5 mL 200 mM Na_2_HPO_4_, pH 3.0) was added to the leaf or leaves. The volume in each well was adjusted to ensure that the leaves were immersed in the DAB solution. The brown precipitate formed by the DAB reacting with the hydrogen peroxide can be observed by light microscopy and photographs are taken by the imaging system. The entire experiments were repeated three times.

### 4.7. Trypan Blue Staining

To study whether *ATL12* was involved with cell death in response to salt stress, over five detached leaves from 4-weeks-old Col-0 wild-type, *atl12* T-DNA insertion mutants, and *ATL12* overexpression lines were treated with 150 mM NaCl for 16 h, no treatment (NT) was used as a control. The treated leaves were then transferred into a 50 mL Falcon tube with a lid and fully immersed with diluted trypan blue solution (10 mL phenol, 10 mL glycerol, 10 mL lactic acid, 10 mL water, and 0.02 g of trypan blue). The tissue was left for 30 min in the staining solution then the staining solution was replaced with 100% ethanol solution. The tissue was then left overnight in the distaining solution and ethanol was replaced several times until the tissue becomes clear. The tissue then can be observed by dissecting microscope and photographs were taken by image system. The experiments were repeated three times.

### 4.8. Water Loss Assay and Stomatal Aperture Measurement

To evaluate the water loss in response to ABA, over five detached leaves from 4-weeks-old plants were obtained and were incubated on 100 μM ABA for 2 h with the abaxial side towards. The leaves incubated with water were used as a control. The leaves’ weight was measured at each indicated time. The water loss was calculated as the loss of the percentage of the fresh weight for each leaf. To measure the stomatal aperture, the method described in Eisele et al., 2016 was used with modification [44]. Over five fresh detached leaves from 4-weeks-old plants were immersed in the stomatal opening solution (30 mM KCl,100 mM CaCl_2_, and 10 mM MES, pH 6.15) for 2 h. Leaves were then cut into half along the midrib. One part of the leaves was incubated with the stomatal opening solution containing ABA for 2 h, and the other part of the leaves was incubated with the stomatal opening solution as a control. All treatments are under constant light conditions. The leaves were then obtained and prepared for scanning electron microscopy. 10–15 stomatal apertures in each leaf were measured. 5 leaves were calculated for each genotype. The images were then measured and analyzed by ImageJ software. The stomatal aperture index (the division of the aperture width through the length) was calculated.

### 4.9. Quantitative Real-Time RT-PCR and RT-PCR

The seedlings with different treatments were collected and frozen in liquid nitrogen and stored in −80 for future qRT-PCR assays. Total RNA was extracted and purified from frozen tissues using TRizol Reagent (Invitrogen^®^, Carlsbad, CA, USA) followed by RNeasy plant mini kit from QIAGEN according to the manufacturer’s protocol. RNA samples were treated with RNase-free DNase I from Bio-Rad (Hercules, CA, USA). First-strand cDNA synthesis was primed with an oligo (dT)15 anchor primer and cDNA was synthesized using the First-Strand Synthesis Kit (Amersham-Pharmacia, Rainham, UK) according to the manufacturer’s protocol. The RT-PCR program consisted of 3 min at 96 °C, 35 cycles of 30 s at 94 °C, 30 s at about 60 °C, and 1 min at 72 °C. The final extension step consisted of 7 min at 72 °C. Amplified PCR fragments were visualized using 1.5% agarose gels. Quantitative RT-PCR experiments were performed using a SYBR^®^ Green qPCR kit (Finnzymes, Espoo, Finland) with reactions at a final volume of 20 µL per well and using the cycle protocol recommended by the manufacturer. Samples were run in the Applied Bioscience qRT-PCR machine. Gene-specific primers were designed using the Geneious software. For quantitative real-time PCR, the following gene-specific primers are used:

*ATL12 (At2g20030):* 5′-GAATTATGCCGTTACTGCGACC-3(forward) and 5′-ATTTTGGCGTGTCGTGTTTAGG-3′(reverse);

*SOS1 (At2g01980):* 5′-GCAAACACTTTGATATTTATCCTCAG-3′(forward) and 5′-CATGAATTCCCTTGGTAGGC-3′(reverse);

*SOS2 (At5g35410):* 5′-CGAGCGAGAAGAATTGAAAGA-3′ (forward), and 5′-CGTTTTGCGGTCTGCTT-3′(reverse);

*AtRBOHD (At5g47910):* 5′-ATCAGTGCCGCATATTCTTTG-3′(forward) and 5′- ATCTTTCTTCCGAAGCACCTC-3′ (reverse);

*AtRBOHF (At1g64060):* 5′- AAACCAACACGCACCTTATTG-3′(forward) and 5′- ATGAAATTGGCATTGCATTTC-3′(reverse);

*RD29B (At5g523000):* 5′-GCAAGCAGAAGAACCAATCA-3′(forward) and 5′-CTTTGGATGCTCCCTTCTCA-3′(reverse);

*CERK1 (At3g21630):* 5′-TCGAAACAGTTCTTGGCGGA-3′(forward) and 5′-GGTTCTCGTCCTGACCCATG-3′(reverse);

*Beta-ACTIN (At3g18780):* 5′-AGCAGCTTCCATTCCCACAA-3(forward) and 5′- CATGCCATCCTCCGTCTTGA -3; (reverse).

Relative fold changes in transcript levels are determined by the double delta Ct Value (ΔΔCt) method. Data were acquired and analyzed using ANOVA followed by Tukey post hoc analysis. Three independent biological replicates will be used in each experiment.

### 4.10. Protein Expression and In Vitro Ubiquitination Assay

ATL12 (At2g20030), UBC8 (At5g41700), UBQ14 (At4g02890) and UBA2 (At5g06460) cDNA were amplified via PCR using primers listed below:

*UBA2* FP: 5′-GGAATTCCATATGATGGAACCATTCGTTGTTAAGG-3′ RP 5′-GGAATTCCTCTCTCTATCTCTGAAACTCA-3′

*UBC8* FP 5′-CGCCATATGATGGCTTCGAAACGGATCTTGAA-3′ RP 5′-CGCGGATCCCTGAAGCATACGAATCTTTGTTTAGC-3′

*UBQ14* FP 5′-GGAATTCCATATGGAATTACAGATGCAGATCTT-3′ RP 5′-CGCGGATGGTTAGAAACCACCACGGAG-3′

*ATL12* FP 5′-GGAATTCCATATGATGAATTCACCACAAGAAATCTCC-3′ RP 5′-CGGATCCCTATACATTAAGATTTTGGCGTGTC-3′

All cDNA of genes were cloned into a modified pET-28a (+) vector containing an N-terminal 6× His tag, which was obtained from Integrated DNA Technologies with the above-mentioned primers. The sequences and direction of all constructs were confirmed by Eurofins Scientific (Luxembourg) DNA tube sequencing. Positive plasmids were transformed into *Escherichia coli* BL21 (DE3) plyss competent cells from ThermoFisher Scientific (Waltham, MA, USA) based on the manufacturer’s protocol, and positive clones were confirmed with plasmid PCR and double restriction enzyme digestion. Two or three fresh positive clones were then picked and incubated with LB medium containing 50 μg/mL kanamycin and 34 μg/mL of chloramphenicol under shaker at 225 rpm overnight at 37 °C. The cells were transferred to 1 L of LB medium containing kanamycin and chloramphenicol in an incubator shaker at 37 °C under 225 rpm until the OD600 reaches about 0.6, BL21 cells were induced with 1 mM isopropyl β-D-1-thiogalactopyranoside (IPTG) overnight at 16 °C. The cells were harvested by centrifugation at 3500 rpm for 10 min and cell pellets were stored at −80 °C for future use. Protein extraction and purification were performed using TALON metal affinity resin with gravity columns kit from Clontech Laboratories, Inc. with modification. 1 g of bacterial cell pellet was suspended in 20 mL xTractor buffer with 40 μL of 5 units/μL DNase I solution, 6 mg lysozyme, 200 μL 0.1 M PMSF, 7 μL BME, and 1 mL of 100 mM MgCl_2_. The cells were lysed with gentle shaking for 30 min at 4 °C and were then centrifuged at 16,000 rpm for 20 min. The supernatant was mixed with 2 mL of equilibrated Talon metal resin with gentle agitation on ice for 20 min. The resin was washed with 40 mL of equilibration buffer containing 300 mM NaCl and 50 mM sodium phosphate at pH 7.4 and then loaded onto the Talon gravity column. Then 40 mL of washing buffer containing 300 mM NaCl, 50 mM sodium phosphate, and 25 mM Imidazole at pH 7.4 was used to wash non-specific binding proteins. The proteins were finally eluted with 10 mL of the elution buffer containing 300 mM NaCl, 50 mM sodium phosphate, and 150 mM Imidazole at pH 7.5 in 500 μL fractions. The protein fraction was determined using a UV spectrometer and SDS-PAGE and then dialyzed with dialysis tubing from ward’s science with 14 kDa molecular weight cut off in 1× phosphate-buffered saline solution contained 137 mM sodium chloride, 2.7 mM potassium chloride, and 10 mM phosphate buffer. The proteins were collected and concentration was determined by Bradford assay (IBI scientific, Dubuque, IA, USA). The proteins were aliquoted in small fractions and stored at −80 °C for downstream application. The in vitro ubiquitination assay was performed with Zhao et al., 2012 protocol with modification. 5× Ubiquitination buffer was prepared containing 100 mM Tris-HCl at pH 7.5, 25 mM MgCl_2_, 2.5 mM DTT and 2 mM ATP. The reactions were prepared in 40 μL total, including 8 μL of 5× ubiquitination buffer, 50 ng of UBA2, 250 ng of UBC8, 500 ng of UBQ14, and 500 ng of ATL12. The reactions minus UBA2, UBC8, UBQ14, ATL12, and ATP, respectively, were prepared at the same time as the control. The reactions were incubated at 30 °C for 2 h. The reactions were stopped by adding 4× SDS sample buffer containing 2 mL 1 M Tris-HCl at pH 6.8, 0.8 g SDS, 4 mL of 100% glycerol, 0.4 mL 14.7 M BME, 1 mL 0.5 M EDTA and 8 mg bromophenol blue, and the samples were boiled at 100 °C for 5 min. The reaction products were separated with 4–20% SDS-PAGE mini protean TGX precast protein gel (Bio-Rad) and detected with the anti-ubiquitin antibody by Western blotting.

## Figures and Tables

**Figure 1 ijms-23-07290-f001:**
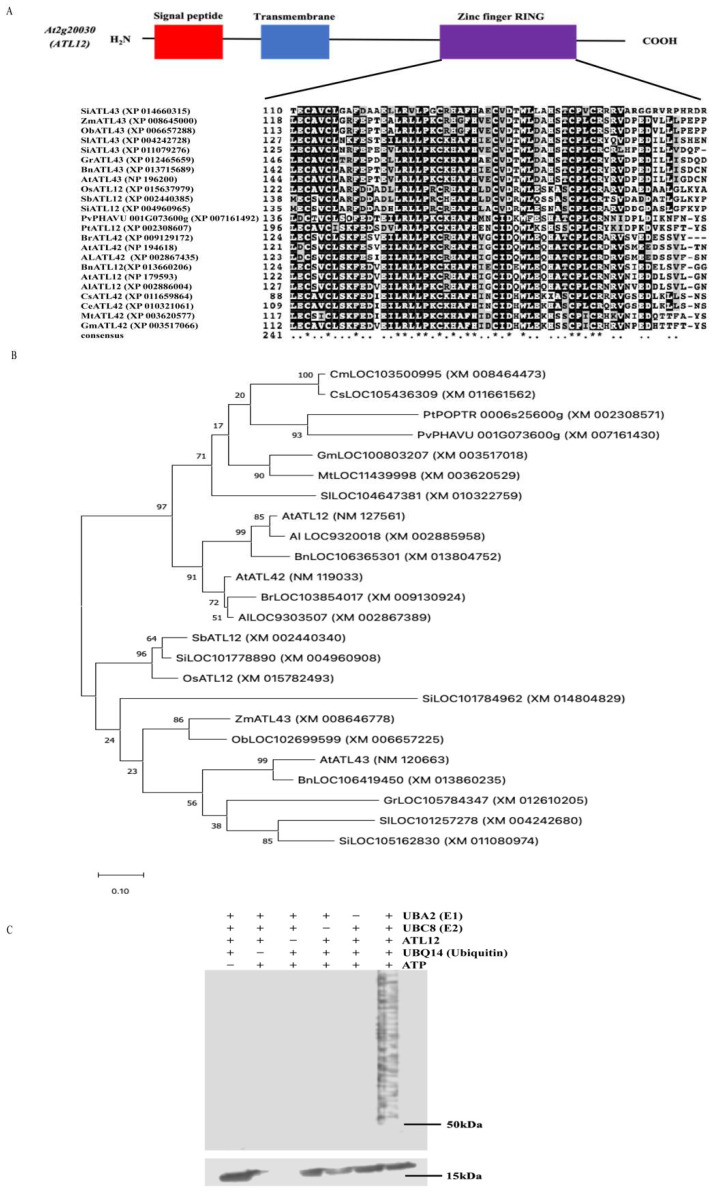
Characterization of ATL12 and the protein sequence and phylogenetic analysis of ATL12. (**A**) Schematic structure of ATL12; Amino acid sequence of ATL12. Red color fonts indicate signal peptide; Blue color fonts indicate transmembrane domain and purple color font show zinc finger RING domain. (**B**) Phylogenetic analysis of the ATL12 homologs. Bootstrap values from 1000 replicates are indicated at each node and the scale represents branch lengths. (**C**) In vitro ubiquitination assay of ATL12. ATL12 was assayed for E3 activity in the presence of E1 (ATL12), E2 (UBC8), and 6× His tagged ubiquitin. Samples were separated by 4–20% SDS-PAGE. Anti-ubiquitin antibody was used to detect in western blotting.

**Figure 2 ijms-23-07290-f002:**
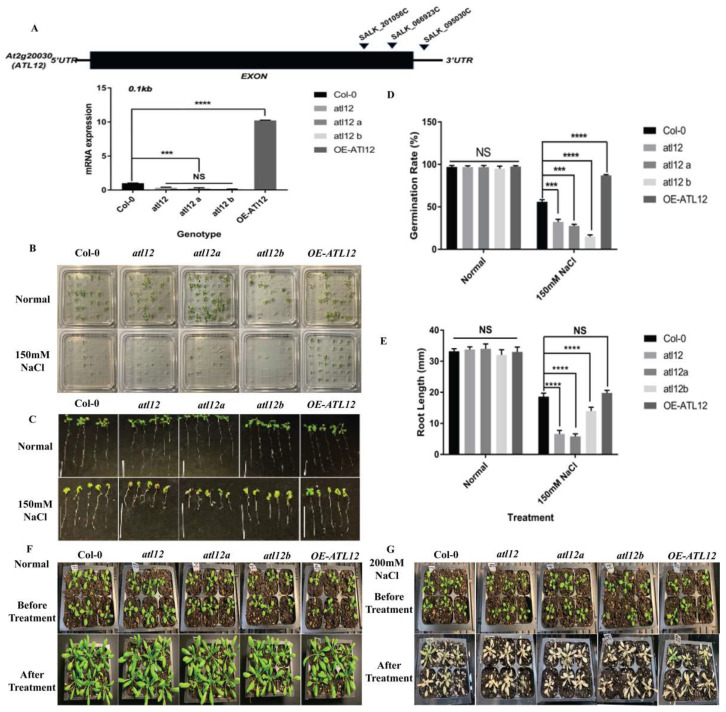
Phenotypic analysis of *ATL12* in response to salt stress. (**A**) Analyses of T-DNA mutants in the *AT2G20030* gene. The locations of the T-DNA insertions in the AT2G20030 gene, *SALK_066923C*, *SALK_095030C*, and *SALK_201056C*, are shown by arrowheads. qRT-PCR analysis of *ATL12* transcript expression in the *atl12* mutant, Col-0 wild type, and *ATL12* over-expression line. Error bars indicate mean ± SD of three independent experiments; (**B**) Images of *Col-0* wild type, *atl12* mutants, and overexpression line germinated in normal and 150 mM NaCl contained MS agar plates. (**C**) Images of *Col-0* wild type, *atl12* mutants, and overexpression line root length growth. (**D**) Germination rate analysis of *Col-0* wild type, T-DNA insertional mutants of *atl12*, *atl12a*, and *atl12b*, and the overexpression (OE) line of *ATL12* in response to salt stress. Error bars indicate mean ± SD of five independent experiments. (**E**) Primary root length growth analysis of Col-0 wild type, T-DNA insertional mutants of *atl12*, *atl12a*, and *atl12b*, and the overexpression (OE) line of *ATL12* in response to salt stress. (**F**,**G**) Salt tolerance of Col-0 wild type, *atl12* mutants, and overexpression line under normal condition and salt condition. NS: no significance; *** indicated *p* < 0.001, **** indicated *p* < 0.0001.

**Figure 3 ijms-23-07290-f003:**
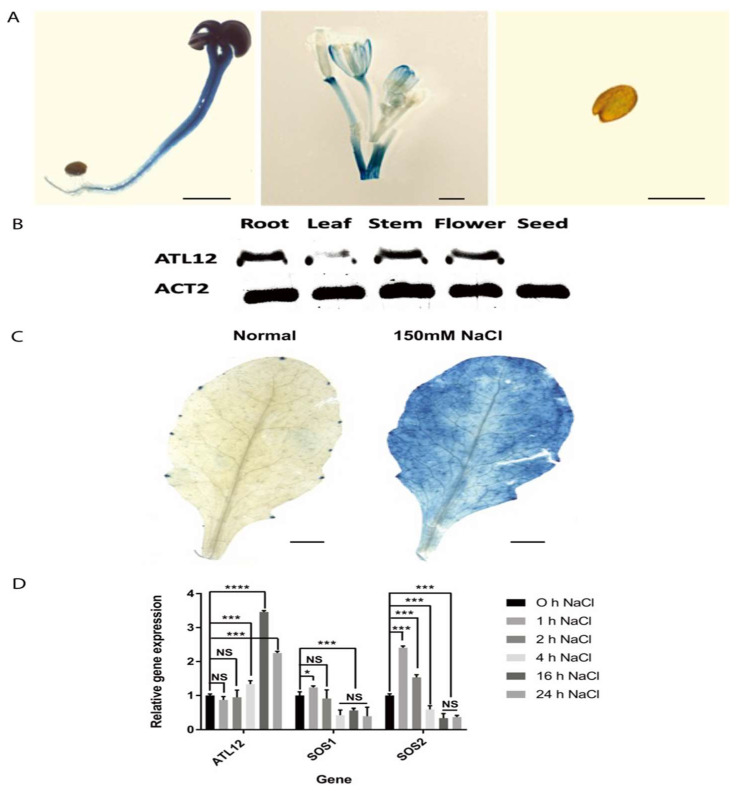
Expression pattern of *ATL12*. (**A**) Histochemical staining analysis of p*ATL12*:GUS seedlings at major organs of *Col-0*. (**B**) RT-PCR analysis of *ATL12* expression in major organs. *Beta-actin(ACT2)* gene expression was used as an internal control. (**C**) Histochemical staining analysis of p*ATL12*:GUS seedlings under normal conditions and 150 mM NaCl condition. (**D**) qRT-PCR analysis of *ATL12* mRNA expression in response to salt at different time points. Asterisks indicate statistically significant differences between the samples treated and untreated, according to the One-way ANOVA analysis and multiple comparisons post-Tukey’s test. The error bar indicates means ± the SD of three independent biological replicates. **** indicates *p* < 0.0001, *** indicates *p* < 0.001, * indicated *p* < 0.05, and NS indicates not significant. The black line indicates that significant differences are present between datasets.

**Figure 4 ijms-23-07290-f004:**
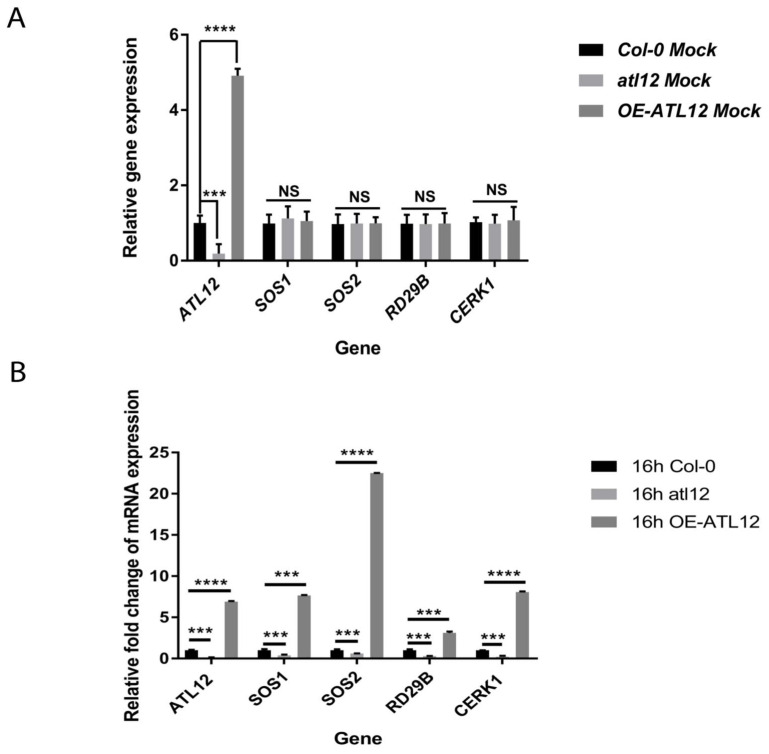
qRT-PCR analysis of *ATL12*, *SOS1*, *SOS2*, *RD29B*, and *CERK1* expression in *Col-0*, *atl12*, and *OE-ATL12* under normal conditions and salt stress. (**A**) qRT-PCR analysis of salt responsive genes under normal conditions; (**B**) qRT-PCR analysis of salt responsive genes in response to salt. Asterisks indicate statistically significant differences between the samples, according to the One-way ANOVA analysis and multiple comparison post Tukey’s test. **** indicates *p* < 0.0001, *** indicates *p* < 0.001, and NS indicates not significant. The black line indicates that significant differences are present between datasets. The error bar indicates means ± the SD of three independent biological replicates.

**Figure 5 ijms-23-07290-f005:**
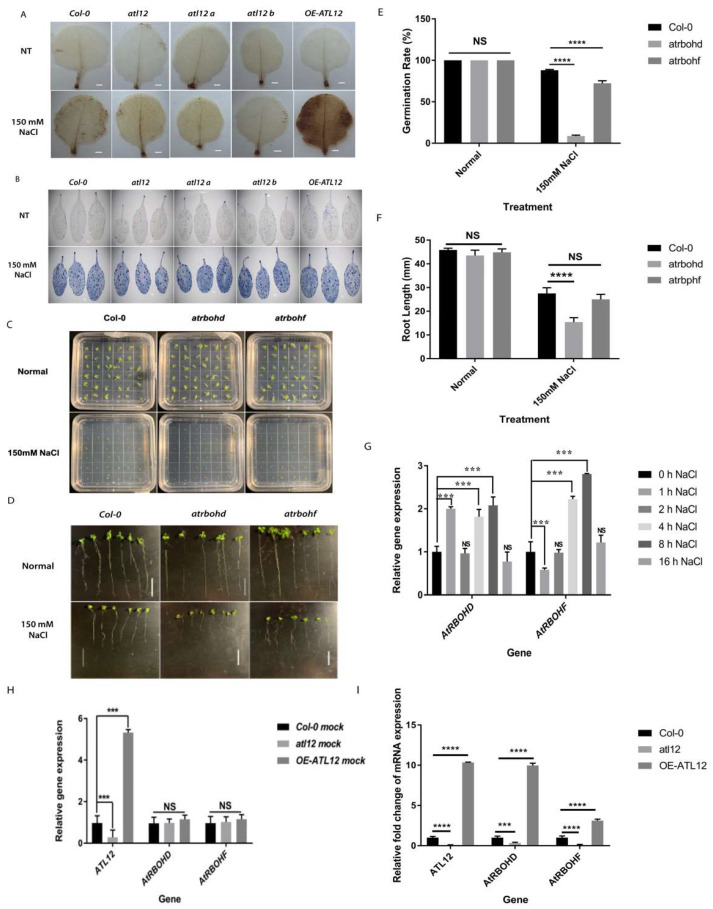
Detection of reactive oxygen species production in over-expression of *ATL12* in response to NaCl stress. (**A**) DAB staining of wild-type, *atl12* mutants and over-expression seedlings in response to salt treatment. DAB staining was performed to indicate areas of hydrogen peroxide production (brown stain indicated the location of hydrogen peroxide activity). The white scale bar indicates 1 mm. (**B**) Trypan blue staining of wild-type, atl12 mutants, and over-expression seedlings in response to salt treatment. (**C**) Germination rate assay in *Col-0*, *atrbohd*, and *atrbohf* in response to 150 mM NaCl. (**D**) Root length growth assay in *Col-0*, *atrbohd*, and *atrbohf* in response to 150 mM NaCl. (**E**) Germination rate analysis of *Col-0* wild type, *atrbohd*, and *atrbohf* mutant in response to salt stress. (**F**) Root length growth of *Col-0* wild-type, *atrbohd*, and *atrbohf* mutant in response to salt stress. Error bars indicate mean ± SD of five independent experiments; NS: no significance; ***: *p* < 0.001, ****: *p* < 0.0001. White scale bar indicates 1.3 cm. (**G**) qRT-PCR analysis of *ATL12*, *AtRBOHD*, and *AtRBOHF* expression in response to salt stress. (**H**) qRT-PCR analysis of *AtRBOHD*, *AtRBOHF*, and *ATL12* expression in mutant of *atl12*, *Col-0* WT, and *ATL12* over-expression line under normal conditions. (**I**) qRT-PCR analysis of *AtRBOHD*, *AtRBOHF*, and *ATL12* expression in mutant of *atl12*, *Col-0* WT, and *ATL12* over-expression line under salt conditions.

**Figure 6 ijms-23-07290-f006:**
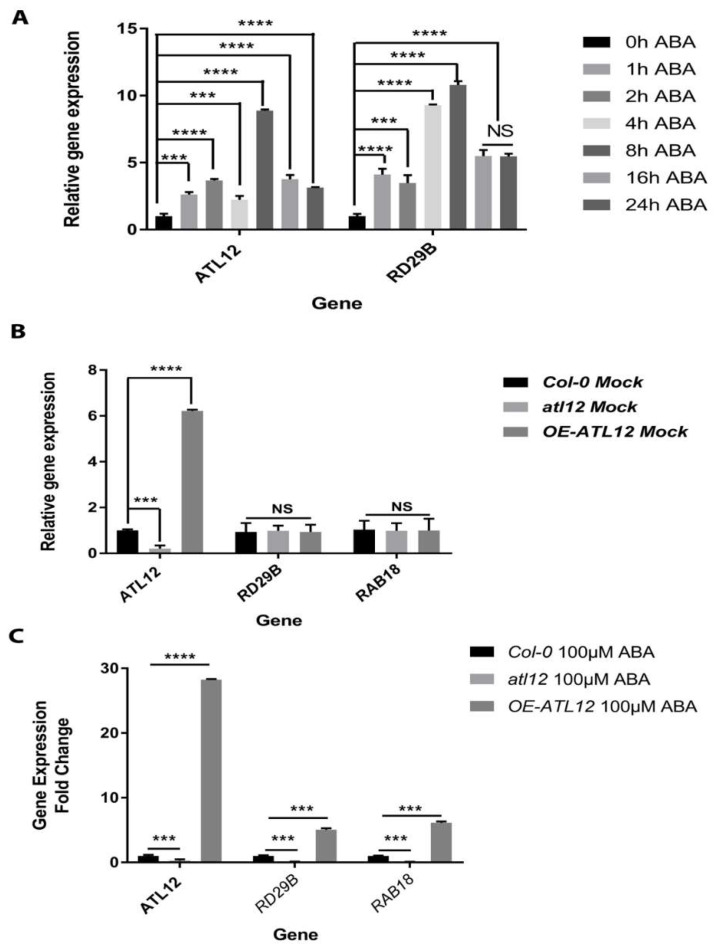
Expression pattern of *ATL12* and *RD29B* in response to ABA. (**A**) qRT-PCR analysis of *ATL12*, and *RD29B* mRNA expression in response to salt at different time points. (**B**) qRT-PCR analysis of ABA-responsive gene *RD29B* and *RAB18* expression in Col-0, *atl12*, and overexpression *ATL12* line under normal conditions in response to ABA. (**C**) qRT-PCR analysis of ABA-responsive gene *RD29B* and *RAB18* expression in Col-0, *atl12*, and overexpression *ATL12* line in response to ABA. Asterisks indicate statistically significant differences between the samples, according to the One-way ANOVA analysis and multiple comparison post-Tukey’s test. NS: no significance; ***: *p* < 0.001, ****: *p* < 0.0001.

**Figure 7 ijms-23-07290-f007:**
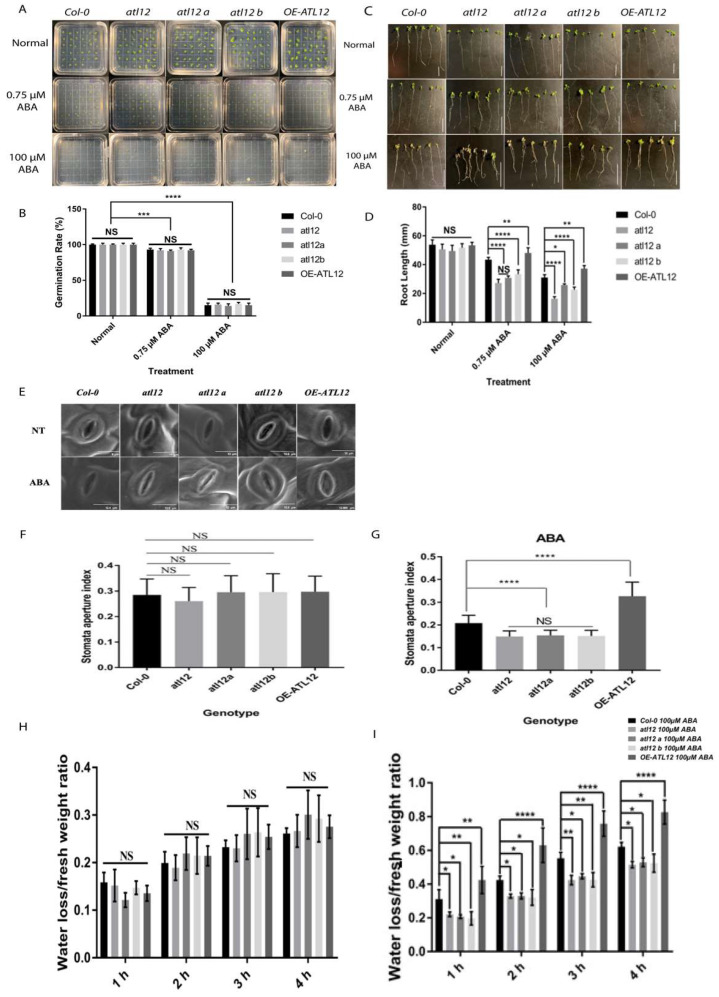
*ATL12* modulates ABA response. (**A**) Images of *Col-0* wild-type, *atl12* mutants, and the overexpression line under normal, 0.75 μM, and 100 μM ABA contained MS agar plates. (**B**) Germination rate analysis. Error bars indicate mean ± SD of five independent experiments; (**C**) Images of *Col-0* wild-type, atl12 mutants, and overexpression line root length growth under normal, 0.75 μM, and 100 μM ABA contained MS agar plates. (**D**) Primary root length analysis. Error bars indicate mean ± SD of five independent experiments; NS: no significance; *: *p* < 0.05, **: *p* < 0.01, ***: *p* < 0.001, ****: *p* < 0.0001. (**E**) SEM image of stomata aperture in *Col-0* wild-type, *atl12* mutants, and overexpression line under normal conditions and ABA treatment. (**F**,**G**) Stomatal aperture assay of *Col-0*, mutants *atl12*, and overexpression line *ATL12* under control and ABA conditions. (**H**) Water loss assay of Col-0, mutants *atl12*, and overexpression line *ATL12* under normal conditions. (**I**) Water loss assay of *Col-0*, mutants *atl12*, and overexpression line *ATL12* under ABA treatment.

**Figure 8 ijms-23-07290-f008:**
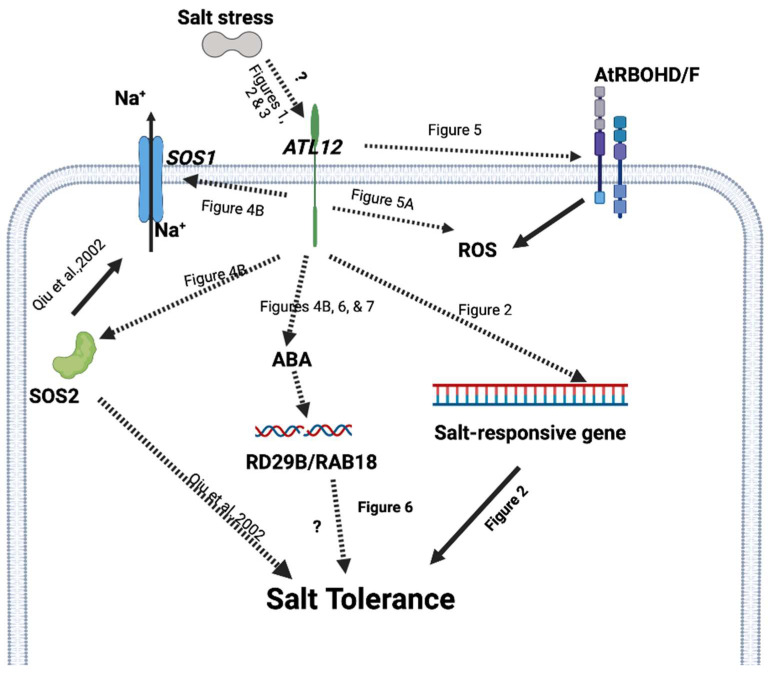
Proposed model for *ATL12*’s function in salt stress response. ATL12 is induced by salt stress and positively regulates salt tolerance in *Arabidopsis* seedlings (Figure 1, Figure 2 and Figure 3). Then *ATL12* modulates salt stress responses via *AtRBOHD/F*-mediated ROS signaling (Figure 5), *SOS1/2* signaling (Figure 4), and ABA-dependent signaling (Figure 6 and Figure 7) in *Arabidopsis thaliana*. A solid arrow indicates confirmed regulation or interaction. Dashed lines indicate our putative interactions with pathways or molecules that are confirmed by results. The question mark indicates further study is required.

## Data Availability

Data is contained within the article or Appendix A.

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
