# Peer review of "Arabidopsis Toxicos en Levadura 12 Modulates Salt Stress and ABA Responses in Arabidopsis thaliana"

_ijms, 2022, doi:10.3390/ijms23137290_

Round 1
Reviewer 1 Report
I find the work well done, organized and straightforward with clear conclusions about the role of ATL12 in regulation of salt stress response. It will be interesting to read the follow-up of the study.
Author Response
Dear Editor and Reviewers,
Thank you for your time and attention in reviewing our paper (manuscript ijms-1765378) and providing valuable comments.
Reviewer 2 Report
The authors have done very nice work and they have experimental data to support their findings. It is a well written manuscript. I have few minor comments for the authors:
1. Please enlarge Figure 2. B-G. Figures are too small and hard to see. Please enlarge Figure 5. C and D. Please enlarge Figure 7. A, C and E.
2. Figure 8 needs revision. Instead of referring to the Figures, please briefly mention the information there. It is not convenient for the readers as it is hard to go back to the figures to understand the whole summary of the mechanism of salt tolerance.
Author Response
Dear Editor and Reviewers,
Thank you for your time and attention in reviewing our paper (manuscript ijms-1765378) and providing valuable comments. We have carefully considered the comments and addressed each one of them. All modifications in the manuscript have been highlighted. Below we provide the point-by-point responses to the reviewers.
1. Please enlarge Figure 2. B-G. Figures are too small and hard to see. Please enlarge Figure 5. C and D. Please enlarge Figure 7. A, C and E.
We reorganized Figures 2, 5 and 7 in order to enlarge them. In addition, we used the Adobe-Ai program to organize our figures, so the zoom-in feature in the figures will not decrease the figure resolution.
2. Figure 8 needs revision. Instead of referring to the Figures, please briefly mention the information there. It is not convenient for the readers as it is hard to go back to the figures to understand the whole summary of the mechanism of salt tolerance.
Thank you for the suggestions. We reorganized Figure 8 and added the brief description as requested.
Reviewer 3 Report
The article presents interesting data about the role of ATL12 in salt stress. The paper is well written, nonetheless several corrections and a careful check is needed:
1. Latin names of the organisms: please use italics all over the manuscript. Be careful, in several cases the second name was written with capital letter (eg. line 126 Glycine Max).
2. Gene and protein names. Full name of genes is not needed to be written in italics (eg. line 430, 441).
3. Abbreviation: eg. WT, OE. Please indicate the meaning of abbreviation when first used.
4. Chemical formula: please correct them, use subscript (small letters for numbers)
5. Use past tense in material and methods (eg. 4.6 and 4.7 subsections)
6. Write the unit of measures consistently, leave a space between the number and the unit of measure (in material and method).
Some of the above mentioned incorrect marking were highighted all over the manuscript. You will find attached.

Author Response
Dear Editor and Reviewers,
Thank you for your time and attention in reviewing our paper (manuscript ijms-1765378) and providing valuable comments. We have carefully considered the comments and addressed each one of them. All modifications in the manuscript have been highlighted. Below we provide the point-by-point responses to the reviewers.
1. Latin names of the organisms: please use italics all over the manuscript. Be careful, in several cases the second name was written with capital letter (eg. line 126 Glycine Max).
Thank you for the comments. We checked the entire text and italiced all organism names.
2. Gene and protein names. Full name of genes is not needed to be written in italics (eg. line 430, 441).
Thank you for the comments. We modified the gene and protein names throughout the text.
3. Abbreviation: eg. WT, OE. Please indicate the meaning of abbreviation when first used.
Thank you for the comments. We indicated the meaning of the abbreviation when first used and made appropriate modifications through the text.
4. Chemical formula: please correct them, use subscript (small letters for numbers)
The chemical formula has been modified in the manuscript.
5. Use past tense in material and methods (eg. 4.6 and 4.7 subsections)
We reorganized the materials and methods in subsections 4.6 and 4.7.
6. Write the unit of measures consistently, leave a space between the number and the unit of measure (in material and method).
Thank you for the comment, we modified the unit of measures through the text.
Sincerely